# Physician Engagement before and during the COVID-19 Pandemic in Thailand

**DOI:** 10.3390/healthcare10081394

**Published:** 2022-07-26

**Authors:** Nantana Suppapitnarm, Montri Saengpattrachai

**Affiliations:** 1Medical Affairs Organization, Bangkok Dusit Medical Services Public Company Limited, Bangkok 10310, Thailand; 2Administrative Office-Chief Medical Officer, Bangkok Hospital Headquarters, Bangkok 10310, Thailand; montri.sa@bangkokhospital.com

**Keywords:** physician engagement, COVID-19, pandemic, BDMS, hospital network, Thailand

## Abstract

The COVID-19 pandemic has affected not only the quality of care and patient safety but also physician engagement. The aim of this study was to investigate physician engagement before and during the COVID-19 pandemic and to identify the areas to improve regarding physician engagement. An online survey was conducted from April 2019 to September 2020 among the physicians of 44 hospitals under the Bangkok Dusit Medical Services Public Company Limited (BDMS) before and during the COVID-19 pandemic. The results were analyzed using an independent T-test and one-way ANOVA to compare the continuous variables across groups. Multiple linear regression was used to identify and adjust the variables to determine the areas for improvement. Among the 10,746 respondents, physician engagement during the COVID-19 pandemic was significantly higher than in the pre-COVID-19 period (4.12 vs. 4.06, *p*-value < 0.001). The top three recommendations to promote physician engagement during the COVID-19 situation comprised (1) marketing (70%), (2) intra-and inter-organizational communication (69%), and (3) the competency of clinical staff (67%). During the COVID-19 pandemic, the positive outcomes toward physician engagement focused on infra-organizational development. These results can be considered in a strategy to optimize physician engagement, which affects the quality of care and patient safety.

## 1. Introduction

In late 2019, an outbreak of a novel coronavirus started and caused a severe acute respiratory syndrome (SARS)-like illness in Wuhan, China [1]. Later, in January 2020, the World Health Organization (WHO) declared the coronavirus disease 2019 (COVID-19) outbreak as it spread rapidly in all regions, causing over 70 million cases worldwide [2].

Regarding Thailand, although it had efficiently coped with such a situation to a certain extent, as of December 2020, there were approximately 10,000 new laboratory-confirmed polymerase chain reaction (PCR)-positive COVID-19 cases reported per day [3,4].

For the first time, in 2022, COVID-19 cases (new cases, severe cases, ventilated cases, and deaths) have all shown a weekly decrease. New laboratory-confirmed (PCR-positive) COVID-19 cases are decreasing by 34% per day compared to the previous week. Following Songkran and other recent holidays, there was no increase in COVID-19 cases, which represented a true decline in community transmission of the Omicron variant-driven COVID-19 fifth wave in Thailand. After this, the average number of probable Antigen Test Kit (ATK)-positive cases reported per day (9994) decreased by 38%. However, reported ATK cases vary greatly, showing increasing and decreasing trends. Combining ATK-probable cases and PCR-confirmed cases gives the ‘total’ daily case count (14,737), which has been approximately halved compared to its value 7 days previously (27,635) [5].

Physician engagement is the process of encouraging physicians to participate in continuously improving care at the patient, organization, and health system levels [6]. Physician engagement is essential in improving care quality, patient safety, and physician satisfaction and retention. However, engaging physicians has been challenging during the COVID-19 pandemic due to burnout caused by heavy workload conditions [7]. Research has shown the prevalence of burnout to be more than 40%, with the highest rates in frontline healthcare providers, such as emergency medicine, primary care, and critical care [8].

Burnout is a stage of emotional, physical, and mental exhaustion caused by chronic workplace stress that has not been successfully managed. It is this process of erosion that leads to negative work outcomes. Recently, researchers have been investigating concepts that contrast with burnout and engagement [9]. A study by Aryatno found that the correlation between work engagement and burnout was significant and negatively correlated [10]. In addition, studies show that healthcare workers experience anxiety, depression, stress, burnout, and less engagement in their workplaces [11,12,13,14,15,16].

Physician engagement is the process that involves physicians in the continual improvement of the quality of care together with a good patient experience [17,18,19]. It is measured by the devotion of the physician toward their organizations with satisfaction for better alignment and quality improvements [18]. An increased physician engagement score reflects the higher success of the operation of the organization as the physicians are committed to helping and sharing responsibility.

Recently, Schaufeli et al. introduced a three-factor model of work engagement: vigor, dedication, and absorption. Vigor refers to high energy levels, mental resilience, persistence, and a willingness to invest effort in one’s work. Dedication is a sense of pride, significance, enthusiasm, inspiration, and challenge. Absorption reflects deep engrossment in one’s work, full concentration, and difficulty detaching oneself from work, whereby time passes quickly [20].

The purposes of this study were: (1) to assess the physician engagement result before and during the COVID-19 pandemic in Thailand and (2) to identify the areas in which improve physician engagement during the COVID-19 situation.

## 2. Materials and Methods

### 2.1. Study Design

This study used a cross-sectional descriptive and comparative design for collecting the data related to physician engagement from medical/dental staff in Bangkok Dusit Medical Services Public Company Limited (BDMS). We hypothesized that the year of the COVID-19 situation would influence physician engagement. The analysis focused on the areas of improvement. The Hospital Director, Chief Medical Officer, and Chief Executive Officer of BDMS approved the research protocol.

### 2.2. Setting

This study analyzed the BDMS physician engagement from April 2019 to September 2020 using an online questionnaire. To cover a nationwide sample from different areas, the physicians and dentists of BDMS, the largest private hospital operator, with a network of 44 hospitals with 12,467 physicians covering every region of Thailand, were recruited in this study. The demographics included the age, position, specialty, years of employment, and status of all medical/dental staff who worked in the 44 hospitals in 6 regions, including (1) the capital city (8 hospitals), (2) the central region (7 hospitals), (3) the western region (5 hospitals), (4) the north/northeastern region (7 hospitals), (5) the eastern region (11 hospitals), and (6) the southern region (6 hospitals).

### 2.3. Participants

The participants were chosen based on their willingness to respond to the survey and whether they met the required criteria as follows: (a) be a doctor or dentist in a hospital of BDMS, (b) no answers with a score of “0”, which means “do not know/does not apply”, in all topics. The study information was given to the participants as a fact sheet via the online survey and informed consent was obtained.

### 2.4. Sample Size

A priori analysis for two independent means, one-way ANOVA and multiple linear regression, was conducted using the statistical power analysis software G * Power (version 3.1) [21]. The sufficient sample size was determined using an alpha of 0.05 and a power of 0.95, and the required sample size was 1084, 360, and 153, respectively. At the end of the survey, 10,836 responses were received. After removing incomplete responses, 10,746 were considered for the data analysis.

### 2.5. Questionnaire

The online questionnaire, named the BDMS Physician Engagement Survey (BDMS-PES) according to the concept of physician engagement [6,22], was developed by the BDMS Medical Staff Organization [23] and the Utrecht Work Engagement Scale (UWES) was developed by Schaufeli et al. [24]. The pilot questionnaire was tested on 60 physicians from two different hospital regions. The final questionnaire comprised 40 questions. For analyzing the questionnaire items, the Kaiser–Meyer–Olkin (KMO) test and Bartlett’s test of sphericity were applied for sampling adequacy. A KMO value of 0.5 and above indicated that the analysis could proceed to exploratory factor analysis. The first 38 questions covered six major issues: A: accessibility, achieving Cronbach’s alpha of 0.843, indicating good internal reliability and consistency [25] (question A1–A3); F: facilities, achieving Cronbach’s alpha of 0.887, indicating good internal reliability and consistency [25] (F4–F10); C: clinical care and support services, achieving Cronbach’s alpha of 0.962, indicating good internal reliability and consistency [25] (C11–C22); Co: communication and feedback, achieving Cronbach’s alpha of 0.842, indicating good internal reliability and consistency [25] (Co23–Co27); M: management and business, achieving Cronbach’s alpha of 0.921, indicating good internal reliability and consistency [25] (M28–M32); and R: relationship with hospital and loyalty, achieving Cronbach’s alpha of 0.941, indicating good internal reliability and consistency [25] (R33–R38). The questionnaires related to work engagement measured vigor (R37), dedication (R34), and absorption (R36). A Likert scale from 0 to 5 was used: score 0 = Don’t Know/Does Not Apply, 1 = Very Poor/Strongly Disagree, 2 = Poor/Disagree, 3 = Fair/Neutral, 4 = Good/Agree, 5 = Very Good/Strongly Agree. There were two open-ended questions: Question 39, “What would you like to see improved in the next 12 months?”, was asked in both surveys, while Question 40, “If you encounter the COVID-19 situation, what do you think the hospital should do?”, was added in the second survey during the COVID-19 pandemic in 2020.

### 2.6. Data Collection

An online version of the survey questionnaire was generated using Google Surveys, to which a link was generated for collecting data. The questionnaires were distributed to the BDMS physicians of 44 hospitals nationwide. Pre-COVID-19 and BDMS-PES questionnaires were distributed on 23 April 2019, and the responses were completed on 31 May 2019. During COVID-19, questionnaires were distributed on 4 August 2020 and the responses were completed on 30 September 2020. Forty-four hospitals were enrolled in the study as an assessment of unsatisfactory data quality resulted in the exclusion of six hospitals from the analysis.

### 2.7. Statistical Analysis

Statistical analyses were performed using the Statistical Package for Social Sciences (SPSS) Version 27.0 for Windows. Qualitative data were presented as absolute numbers and percentages. Quantitative data were presented as means and standard deviations. Means were compared using an independent T-test and one-way ANOVA. A *p*-value less than 0.05 was considered significant. The significant variables were categorized according to (1) year of survey; (2) region of hospital; (3) gender; (4) age group; (5) years of employment; (6) physician status; (7) specialty; and (8) job position. Multivariate analysis using linear regression analysis was performed to identify and adjust for a range of covariates, including the significant variables from Table 1, which are the factors associated with physician engagement obtained from the literature review, including hospital region, gender, age, years of employment, physician status, and job position [26,27,28,29]. Missing data were removed to avoid any bias in analyzing the results.

## 3. Results

A total of 12,467 physicians were approached to participate in the study, of whom 10,746 physicians (5409 males, 5337 females) completed the survey, yielding a recruitment rate of 86.2%. Table 1 summarizes the demographic backgrounds of the respondents. There were 5294 and 5452 respondents for the surveys before and during the COVID-19 pandemic, respectively. The sex ratio was close to 1:1 in both surveys. The regions of hospitals showed similar percentages between the two surveys. There was no difference in the majority characteristics between the two surveys: age range of 31 to 40 years old, years of employment of 1–5 years, and a ratio of full-time to part-time physicians of 1:1. Most respondents (90%) worked as practitioners only, and the remaining 10% worked as practitioners together with management physicians. The top five specialties among survey participants were Medicine (42.2%), Surgery (27.6%), Pediatrics (10.8%), Dentistry (7.0%), and Obstetrics (6.5%).

The overall physician engagement scores between the periods before and during the COVID-19 pandemic revealed 4.06 versus 4.12, with a statistically significant *p*-value < 0.001. Among the respondent characteristics, seven variables showed significant differences (*p*-value < 0.001) among the physician engagement scores between groups: (1) year of survey, (2) region of hospital, (3) gender, (4) age group, (5) years of employment, (6) physician status, and (7) job position. The specialty of respondents did not show a statistical difference in physician engagement scores (*p*-value 0.173), as presented in Table 2.

Table 3 shows the multiple linear regression analysis to predict physician engagement based on the factors involved, controlling for potential confounders. A significant regression equation was found (F (28, 10,565) = 222,981.53, *p* value < 0.001), with an R2 of 0.998. The predicted physician engagement result is equal to 0.002 + 0.061 (year of COVID-19) + 0.082 (accessibility) + 0.17 (facilities) + 0.302 (clinical care and support services) + 0.139 (communication and feedback) + 0.137 (management and business) + 0.169 (relationship with hospital and loyalty) −0.003 (hospital group 3 or hospital in western region). Clinical care and support services was found to be the strongest predictor of the physician engagement result (*p* < 0.001), followed by clinical care and support services (*p* < 0.001), relationship with hospital and loyalty (*p* < 0.001), communication and feedback (*p* < 0.001), management and business (*p* < 0.001), facilities (*p* < 0.001), accessibility (*p* < 0.001), year of COVID-19 (*p* < 0.001), and hospital in western region (*p =* 0.003).

Table 4 compares physician engagement scores on 38 questions between the periods before during the COVID-19 pandemic using multivariate analysis. The majority of the scores (34 out of 38 questions) were significantly associated with the year of survey differences (*p*-value < 0.05). Only four questions, including A1—scheduling process responsive and appropriate, C21—accounting and finance service, R33—hospital delivers on its promises, and R36—it is difficult to detach myself from my work, were not significantly associated with the year of the survey (*p*-value 0.476, 0.063, 0.739, 0.931, respectively). For work engagement, vigor and dedication were significantly different and associated with the year of the survey (*p*-value < 0.05), while absorption was not significant (*p*-value = 0.931).

Figure 1 shows the areas of improvement suggested by the respondents. Pre-COVID-19, the top three topics that physicians would like to see improved were doctor fees (72.5%), doctor accommodation/doctor lounge (72.2%), and car park (69.8%). Meanwhile, during COVID-19, the top three topics were marketing (70%) followed by intra-organizational communication (69%) and competency of clinical staff (67%).

Question 40 was added in the second survey during the COVID-19 pandemic in 2020: “If you encounter the COVID-19 situation, what do you think the hospital should do?” The most common recommendations made by the respondents included promoting telemedicine and teleconsultation to patients, providing health literacy and knowledge about COVID-19 infection to the public, and encouraging people to follow universal prevention measures such as social distancing and handwashing.

## 4. Discussion

In 2020, the entire world was confronted with a difficult situation owing to the COVID-19 outbreak, which has affected nations worldwide. Business operations have been abruptly disrupted, with no exception for the healthcare system. According to the Bureau of Labor Statistics, the U.S. healthcare sector has lost nearly half a million workers since February 2020, and approximately 1 in 5 healthcare workers, or 18%, have quit their jobs since the pandemic began. The prevalence of physician burnout has increased to 68% during the pandemic [30]. These have caused a broad impact in which we have all been unavoidably affected due to the crisis, particularly healthcare providers in both public and private hospitals. This study compared physician engagement before and during the COVID-19 pandemic in Thailand and focused on the areas of improvement that positively influence physician engagement to maintain doctors within their organizations. Studies in the public sector showed that healthcare workers were dissatisfied with their jobs during the COVID-19 pandemic [31]. However, very few studies have been performed in the private sector.

“Physician engagement” is a commonly used term in healthcare management. It refers to physicians who are committed to the organization’s mission and are willing to help the organization when required [18]. In Thailand, a physician engagement survey among 44 BMDS hospital members has been carried out since 2012. This study showed that physician engagement was affected during the COVID-19 pandemic. In contrast to many studies that showed deteriorated results due to having experienced fear, panic, anxiety, depression, and burnout among healthcare workers [13,16,32,33,34], the physician engagement in our survey showed a higher score during the COVID-19 pandemic (4.12 versus 4.06, *p*-value < 0.001). Hospital management is the key to the success of physician engagement, which has been mentioned in some studies [35,36]. Management is defined as a process comprising social and technical functions, and activities in organizations to accomplish predetermined objectives through humans and other resources. The management team is the key to the success of management because the management team needs to support and coordinate the services provided within healthcare organizations [37]. Good hospital management could overcome fear, strengthen engagement, and eventually improve hospital performance and contribute to a sustainable workplace. As part of management, BDMS considers physician engagement as one of the flagships in developing loyalty amongst physicians and dentists towards the organization. Since 2012, physician engagement has been surveyed throughout the BDMS hospital network. The result has been utilized for developing the continuous quality improvement (CQI) project every year. This could be the reason that the organizational loyalty amongst physicians and dentists was maintained even during the crisis. It is challenging to compare our results with the others because in-depth information on the physician engagement-related strategies of each organization is limited.

The sociodemographic characteristics, such as hospital group, gender, age group, status group, and job position, significantly influenced physician engagement (*p*-value < 0.001). These findings are similar to the study performed by Yong Lu et al. [38], where they found that among the sociodemographic variables, occupation, educational background, professional status, years of service, annual income, and night shift frequency significantly influenced the level of job satisfaction. Another study by Feng Jiang et al. [39] mentioned that differences in gender and the region of respondents explicitly impacted the satisfaction scores.

Regarding the specialty of the respondents, there was no impact on physician engagement. However, from our viewpoint, the art of administration of physicians should be harmonized and not influenced by the expertise of the physicians. These findings were supported by Nunez-Smith et al. [40], who mentioned that discrimination remains a problem for the medical profession, threatening efforts at creating a physician workforce that reflects the diversity of the American people. Developing and retaining a diverse physician workforce will require the active engagement of all physicians and healthcare organizations at every level of the healthcare system.

Considering the survey topics in Table 3, three topics had lower scores during the COVID-19 pandemic and reflected no improvement plan for a long time: A1 “scheduling process”, R33 “hospital delivers on its promises, and R36 “it is difficult to detach myself from my work”. These topics should be considered for improvement.

Regarding work engagement, our study found higher scores for vigor and dedication during the COVID-19 survey. The reason is that the physicians are more aware and committed to their work. A full-time physician, according to the BDMS Physician Bylaws, works at least 40 h per week and is not allowed to leave the job. There is also an opportunity for doctors to participate as volunteers in the care of COVID-19 patients. Doctors may experience pride in helping people during epidemic situations. On the other hand, the absorption is no different because the workload increases the burden of responsibility, which may make it difficult to separate, thus making this part no different.

Regarding what physicians would like to see improved in the next 12 months, as shown in Figure 1, the top three recommendations for enhancing physician engagement were shifted from doctor income, e.g., doctor fees, doctor accommodation/doctor lounge, and car park to marketing, intra- and inter-organizational communication, and the competency of the staff during the COVID-19 pandemic. Each recommendation is detailed as follows: Marketing: in addition to the government policy to implement an Acute Respiratory Infection (ARI) clinic nationwide, the BDMS should highlight the newly introduced services to the public, such as the teleconsultation services, together with e-medical treatment, medication home delivery services, drive-through COVID-19 tests, etc.Intra- and inter-organizational communication: this is a crucial topic that needs to be addressed, especially regarding up-to-date information or even the professional standard-related incidence. Communication with doctors can be either official or unofficial and can take place through various channels. This physician recommendation is similar to the study of Matthew A Crain et al. [41].Competency of clinical staff: upskilling and reskilling of clinical staff working with physicians, e.g., nurses, should be consistently provided by the hospital. The organization needs to offer wide-ranging and easily accessible learning platforms, such as e-learning, in-house academic meetings, etc. Physicians also need to update their competencies during the uncertain and unpredictable COVID-19 situation. This recommendation aligns with other studies worldwide [26,42].

## 5. Limitations

Data from physicians in the government sector and on specific specialties, such as critical care doctors, are limited in this study and should be considered in the future.

## 6. Implications of the Study

Physician engagement is critical for improving efficiency, quality of care, patient safety, and physician satisfaction and retention [22]. Many hospitals still struggle to improve physician engagement, especially during the COVID-19 pandemic. This paper sheds light upon physicians’ insights and identifies areas for improvement.

This study highlights the higher engagement during the COVID-19 pandemic. Cooperation between the management team and physicians at the highest level and inter-professional communication are the keys to success, rather than the incentive and facility factors. For hospital-wide survival, cost-saving strategies by reducing doctors’ salaries but not laying off doctors are deployed. Some parts of salaries were substituted by wellness programs, COVID-related health insurance for doctors, and vaccines against COVID-19. Moreover, using technology to support work engagement during COVID-19 is very helpful to physicians and healthcare teams—for example, the use of telemedicine technology to create public relations media for patients and healthcare staff to understand the universal precautions; self-protection measures such as handwashing and using personal protection equipment (PPE); and teleconsultation to treat patients who are not able to attend the hospital during COVID-19 or who are admitted in the hospital field.

As a result, many hospitals can adopt physician engagement as a flagship strategic priority to improve healthcare overall. It is critical for hospital administrators and physician leadership to develop and utilize relevant skills to enhance engagement levels.

## 7. Conclusions

Physician engagement is critical for a successful patient care experience, especially during a challenging situation such as COVID-19. Physician engagement showed a higher score during the COVID-19 pandemic. The data from this study can help the hospital management team to develop a continuous quality improvement project to increase physician engagement. High levels of physician engagement influence the healthcare workforce and drive organizational strategies and development. This information could lead to a positive impact on the quality of patient care.

## Figures and Tables

**Figure 1 healthcare-10-01394-f001:**
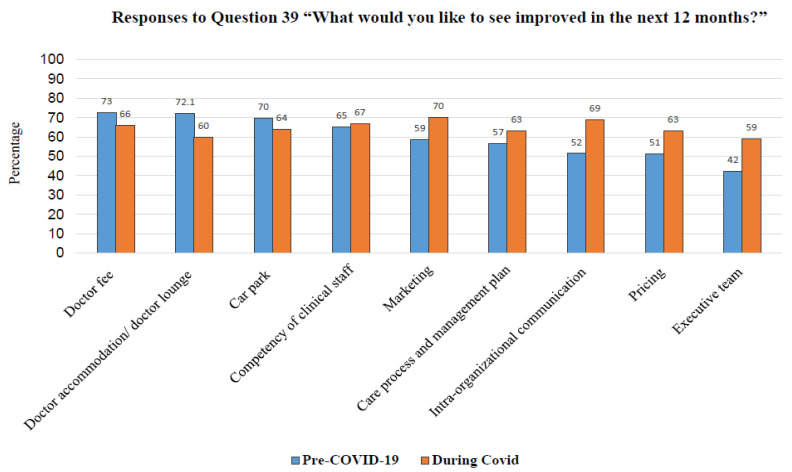
Responses to Question 39, “What would you like to see improved in the next 12 months?”.

**Table 1 healthcare-10-01394-t001:** The characteristics of medical/dental staff who responded to the engagement surveys.

Respondent Characteristics		Total (*n* = 10,746)	Before COVID-19 (*n =* 5294)	During COVID-19 (*n =* 5452)
Hospital’s Region	Capital City (8 hospitals)	1715 (16.0%)	814 (15.4%)	901 (16.5%)
	Central (7 hospitals)	2558 (23.8%)	1288 (24.3%)	1270 (23.3%)
	Western (5 hospitals)	1111 (10.3%)	508 (9.6%)	603 (11.1%)
	North/Northeastern (7 hospitals)	1378 (12.8%)	675 (12.8%)	703 (12.9%)
	Eastern (11 hospitals)	2929 (27.3%)	1478 (27.9%)	1451 (26.6%)
	Southern (6 hospitals)	1055 (9.8%)	531 (10.1%)	524 (9.6%)
Gender	Male	5409 (50.3%)	2691 (50.8%)	2718 (49.9%)
	Female	5337 (49.7%)	2603 (49.2%)	2734 (50.1%)
Age Group	20–30 years	761 (7.1%)	423 (8.0%)	338 (6.2%)
	31–40 years	5311 (49.4%)	2557 (48.3%)	2754 (50.5%)
	41–50 years	2564 (23.9%)	1237 (23.4%)	1327 (24.3%)
	51–60 years	1260 (11.7%)	648 (12.2%)	612 (11.2%)
	>60 years	850 (7.9%)	429 (8.1%)	421 (7.7%)
Years of Employment	0–11 months	1406 (13.1%)	834 (15.8%)	572 (10.5%)
	1–5 years	4634 (43.1%)	2070 (39.1%)	2564 (47.0%)
	6–10 years	2261 (21.0%)	1145 (21.6%)	1116 (20.5%)
	11–15 years	1132 (10.5%)	586 (11.1%)	546 (10.0%)
	16–20 years	531 (4.9%)	245 (4.6%)	286 (5.2%)
	21–25 years	358 (3.3%)	185 (3.5%)	173 (3.2%)
	26–30 years	200 (1.9%)	111 (2.1%)	89 (1.6%)
	>30 years	224 (2.1%)	118 (2.2%)	106 (1.9%)
Physician Status	Full Time	4652 (43.3%)	2309 (43.6%)	2343 (43.0%)
	Part Time	6094 (56.7%)	2985 (56.4%)	3109 (57.0%)
Specialty Group	Medicine	4531 (42.2%)	2301 (43.5%)	2230 (40.9%)
	Surgery	2969 (27.6%)	1436 (27.1%)	1533 (28.1%)
	Obstetrics	696 (6.5%)	342 (6.5%)	354 (6.5%)
	Pediatrics	1161 (10.8%)	583 (11.0%)	578 (10.6%)
	Radiology	528 (4.9%)	263 (4.9%)	265 (4.9%)
	Dentistry	755 (7.0%)	326 (6.2%)	429 (7.9%)
	General	106 (1.0%)	43 (0.8%)	63 (1.2%)
Job Position	Practice Only	9668 (90.0%)	4743 (89.6%)	4925 (90.3%)
	Director of Department	442 (4.1%)	239 (4.5%)	203 (3.7%)
	Management	331 (3.1%)	174 (3.3%)	157 (2.9%)
	Not Specified	305 (2.8%)	138 (2.6%)	167 (3.1%)

**Table 2 healthcare-10-01394-t002:** The study variables and physician engagement scores.

Study Variables		*n*	Physician Engagement Score	t/F	*p*-Value
			Mean ± SD		
Year of Survey	Before COVID-19	5294	4.06 ± 0.51	t = −5.624	<0.001 ^a^
	During COVID-19	5452	4.12 ± 0.54		
Hospital’s Region	Capital City (8 hospitals)	1715	4.07 ± 0.52	F = 32.919	<0.001 ^b^
	Central (7 hospitals)	2558	4.13 ± 0.51		
	Western (5 hospitals)	1111	4.02 ± 0.55		
	North/Northeastern (7 hospitals)	1378	4.22 ± 0.52		
	Eastern (11 hospitals)	2929	4.02 ± 0.54		
	Southern (6 hospitals)	1055	4.09 ± 0.50		
Gender	Male	5409	4.12 ± 0.54	t = 5.982	<0.001 ^a^
	Female	5337	4.06 ± 0.52		
Age Group	20–30 years	761	4.19 ± 0.53	F = 23.217	<0.001 ^b^
	31–40 years	5311	4.12 ± 0.53		
	41–50 years	2564	4.04 ± 0.54		
	51–60 years	1260	4.02 ± 0.52		
	>60 years	850	4.05 ± 0.49		
Years of Employment	0–11 months	1406	4.17 ± 0.50	F = 21.869	<0.001 ^b^
	1–5 years	4634	4.13 ± 0.52		
	6–10 years	2261	4.05 ± 0.55		
	11–15 years	1132	4.00 ± 0.54		
	16–20 years	531	3.96 ± 0.55		
	21–25 years	358	4.01 ± 0.46		
	26–30 years	200	4.00 ± 0.52		
	>30 years	224	4.04 ± 0.53		
Physician Status	Full Time	4652	3.98 ± 0.54	t = −18.791	<0.001 ^a^
	Part Time	6094	4.17 ± 0.51		
Specialty Group	Medicine	4531	4.10 ± 0.52	F = 1.502	0.173 ^b^
	Surgery	2969	4.07 ± 0.54		
	Obstetrics	696	4.09 ± 0.51		
	Pediatrics	1161	4.09 ± 0.53		
	Radiology	528	4.07 ± 0.51		
	Dentistry	755	4.07 ± 0.51		
	General	106	4.15 ± 0.62		
Job Position	Practice Only	9668	4.08 ± 0.53	F = 10.113	<0.001 ^b^
	Director of Department	442	4.07 ± 0.45		
	Management	331	4.07 ± 0.48		
	Not Specified	305	4.25 ± 0.61		

Note: ^a^: independent T-test, ^b^: one-way ANOVA.

**Table 3 healthcare-10-01394-t003:** Multiple linear regression analysis to predict physician engagement based on the factors involved.

Model	Unstandardized Coefficients		Standardized Coefficients			95.0% Confidence Interval for B	
	B	Standard Error	Beta	t	*p*-Value	Lower Bound	Upper Bound
(Constant)	0.002	0.003		0.740	0.460	−0.003	0.007
Year of Survey (0 = Before COVID-19, 1 = During COVID-19)	0.061	0.015	0.039	4.078	0.000	0.032	0.090
Gender (0 = Male, 1 = Female)	−0.001	0.000	−0.001	−1.287	0.198	−0.001	0.000
Status (0 = Full Time, 1 = Part Time)	−0.001	0.000	−0.001	−1.830	0.067	−0.002	0.000
Accessibility Score	0.082	0.000	0.096	168.260	0.000	0.081	0.083
Facilities Score	0.170	0.001	0.193	300.173	0.000	0.169	0.171
Clinical Care Score	0.302	0.001	0.320	428.396	0.000	0.301	0.304
Communication Score	0.139	0.001	0.164	214.801	0.000	0.138	0.141
Management Score	0.137	0.001	0.180	241.365	0.000	0.136	0.138
Relationship Score	0.169	0.001	0.204	322.978	0.000	0.168	0.170
Hospital Region_ Capital	−0.001	0.001	−0.001	−1.544	0.123	−0.003	0.000
Hospital Region_Central	5.726 × 10^−5^	0.001	0.000	0.070	0.944	−0.002	0.002
Hospital Region_Western	−0.003	0.001	−0.002	−3.018	0.003	−0.005	−0.001
Hospital Region_North/Northern	−0.001	0.001	0.000	−0.845	0.397	−0.003	0.001
Hospital Group_Eastern	−0.001	0.001	−0.001	−1.835	0.067	−0.003	0.000
Age Group 20–30	0.001	0.001	0.001	0.947	0.344	−0.001	0.004
Age Group 31–40	0.001	0.001	0.001	1.287	0.198	−0.001	0.003
Age Group 41–50	0.000	0.001	0.000	0.327	0.744	−0.002	0.002
Age Group 51–60	0.001	0.001	0.001	1.274	0.203	−0.001	0.003
Years of Employment < 1 year	−0.001	0.002	−0.001	−0.701	0.483	−0.005	0.002
Years of Employment 1–5	0.000	0.002	0.000	0.211	0.833	−0.003	0.004
Years of Employment 6–10	0.000	0.002	0.000	0.59	0.953	−0.003	0.003
Years of Employment 11–15	−0.001	0.002	−0.001	−0.628	0.530	−0.005	0.002
Years of Employment 15–20	0.000	0.002	0.000	−0.119	0.905	−0.004	0.003
Years of Employment 21–25	0.001	0.002	0.000	0.696	0.486	−0.002	0.005
Years of Employment 25–30	−0.001	0.002	0.000	−0.694	0.488	−0.006	0.003
Job Position_Practice Only	−0.003	0.001	−0.001	=1.952	0.051	−0.005	0.000
Job Position_Director of Department	−0.003	0.002	−0.001	−1.664	0.096	−0.006	0.000
Job Position_Management Only	−0.003	0.002	−0.001	=1.436	0.151	−0.06	0.001

Note: multiple linear regression R = 0.999, R^2^ = 0.998, F = 222,981.531, *p*-value < 0.00.

**Table 4 healthcare-10-01394-t004:** Physician engagement scores categorized by survey topic.

Survey Topics	Engagement Score		Unadjusted Mean Diff (95% CI)	Adjusted Mean		*p*-Value
	(Mean ± SD)			Diff (95% CI)		
	Pre-COVID-19	During COVID-19		Lower	Upper	
**A: Accessibility**						
A1: Scheduling process responsive and appropriate	4.21 ± 0.696	4.20 ± 0.714	−0.010	−0.037	0.017	0.476
A2: Access to and availability of patient record	4.09 ± 0.736	4.13 ± 0.739	0.028	0.000	0.056	0.049
A3: Ambulatory services	4.16 ± 0.689	4.20 ± 0.688	0.048	0.022	0.075	<0.001
**F** **: Facilities**						
F4: On-call doctor accommodation	3.92 ± 0.859	4.00 ± 0.831	0.091	0.055	0.127	<0.001
F5: Doctor lounge	4.02 ± 0.801	4.10 ± 0.797	0.082	0.051	0.113	<0.001
F6: Medical examination room	4.09 ± 0.733	4.16 ± 0.723	0.065	0.037	0.093	<0.001
F7: Operating room	4.13 ± 0.682	4.24 ± 0.681	0.097	0.064	0.13	<0.001
F8: Availability of preferred equipment	4.04 ± 0.728	4.12 ± 0.735	0.074	0.046	0.101	<0.001
F9: Cleanliness of facilities	4.21 ± 0.718	4.30 ± 0.744	0.085	0.058	0.112	<0.001
F10: Quality of food and cleanliness	3.86 ± 0.841	3.93 ± 0.858	0.079	0.046	0.112	<0.001
**C: Clinical care and support services**						
C11: On-call doctors are good	4.00 ± 0.689	4.08 ± 0.689	0.077	0.048	0.106	<0.001
C12: Nursing service	4.09 ± 0.690	4.14 ± 0.693	0.057	0.031	0.084	<0.001
C13: Pharmacy service	4.20 ± 0.639	4.27 ± 0.638	0.073	0.049	0.098	<0.001
C14: Radiology service	4.18 ± 0.626	4.25 ± 0.632	0.075	0.05	0.099	<0.001
C15: Laboratory service	4.13 ± 0.643	4.19 ± 0.663	0.061	0.036	0.086	<0.001
C16: Information technology service	3.98 ± 0.752	4.06 ± 0.745	0.078	0.049	0.107	<0.001
C17: Biomedical engineering service	4.05 ± 0.677	4.13 ± 0.685	0.081	0.054	0.107	<0.001
C18: Reception service	4.21 ± 0.634	4.27 ± 0.656	0.056	0.031	0.081	<0.001
C19: Referral center service	4.09 ± 0.662	4.16 ± 0.664	0.069	0.042	0.097	<0.001
C20: Marketing service	3.89 ± 0.815	3.95 ± 0.844	0.060	0.026	0.094	<0.001
C21: Accounting and finance service	4.12 ± 0.664	4.15 ± 0.707	0.025	−0.001	0.052	0.063
C22: Teamwork among care team	4.08 ± 0.722	4.14 ± 0.738	0.061	0.034	0.089	<0.001
**Co: Communication and feedback**						
Co23: The ability of hospital staff to respond and accurately resolve issues.	3.92 ± 0.726	4.00 ± 0.740	0.080	0.053	0.107	<0.001
Co24: I have the opportunity to review this hospital’s patient satisfaction data.	3.88 ± 0.760	3.98 ± 0.758	0.100	0.069	0.130	<0.001
Co25: I am satisfied with the communication I receive from the clinical staff about my patients.	4.01 ± 0.691	4.08 ± 0.708	0.070	0.044	0.097	<0.001
Co26: My orders are carried out to my satisfaction.	4.06 ± 0.688	4.11 ± 0.723	0.053	0.026	0.079	<0.001
Co27: The hospital provides high-quality care and services.	4.16 ± 0.682	4.21 ± 0.699	0.054	0.028	0.080	<0.001
**M: Management and business**						
M28: Hospital information readily available to doctor.	3.93 ± 0.742	4.01 ± 0.753	0.088	0.06	0.117	<0.001
M29: Hospital support and responsiveness to doctors’ needs.	3.92 ± 0.788	3.98 ± 0.806	0.058	0.029	0.088	<0.001
M30: Opportunity for giving opinions in hospital work.	3.85 ± 0.824	3.89 ± 0.857	0.051	0.018	0.084	0.002
M31: Hospital provides continuing medical education for physicians to develop an excellent healthcare center.	3.95 ± 0.806	4.03 ± 0.814	0.077	0.045	0.109	<0.001
M32: Overall, how satisfied are you with the management/running of the hospital?	4.03 ± 0.738	4.08 ± 0.767	0.052	0.024	0.081	<0.001
**R: Relationship with hospital and loyalty**						
R33: Hospital delivers on its promises.	4.04 ± 0.730	4.03 ± 0.789	−0.005	−0.034	0.024	0.739
R34: I am proud to work with the hospital and I am a part of this organization (dedication).	4.24 ± 0.686	4.27 ± 0.705	0.038	0.012	0.064	0.005
R35: Hospital treats me with respect.	4.25 ± 0.727	4.27 ± 0.754	0.029	0.001	0.057	0.041
R36: It is difficult to detach myself from my work (absorption).	3.97 ± 0.792	3.96 ± 0.842	−0.001	−0.033	0.030	0.931
R37: When I get up in the morning, I feel like going to work (vigor).	4.07 ± 0.732	4.14 ± 0.736	0.081	0.053	0.109	<0.001
R38: I would recommend other doctors to work with this hospital.	4.13 ± 0.762	4.19 ± 0.782	0.061	0.032	0.090	<0.001

Note: multiple linear regression. Adjusted for year of survey, hospital region, gender, age, years of employment, physician status, and job position. CI: confidence interval.

## Data Availability

The data are not publicly available due to containing information that could compromise the privacy of research participants.

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
