# Peer review of "Physician Engagement before and during the COVID-19 Pandemic in Thailand"

_healthcare, 2022, doi:10.3390/healthcare10081394_

Round 1
Reviewer 1 Report
Many thanks for the opportunity to review this interesting study. Overall, the study is well written but can be improved further. Please see the comments below for your consideration:
1. The introduction needs some attention to present more information regarding physician engagement and its determinants as noted in existing literature. The information provided so far is quite limited. The paragraph from line 43 to 54 can be restructured to present the definition of physician engagement before the details regarding burnout.
2. Regarding materials and methods, there is no section for study design. Instead, it has been captured under data collection. It will be helpful if this is made explicit. Also, since this is cross-sectional study, it will be helpful if the authors use the STROBE checklist in reporting the study. In fact, there is so much information packed under the data collection section (such as participant recruitment, sample size etc). This is rather confusing and should be corrected. Hopefully if the authors are to follow the STROBE guidelines, it will be possible to provide appropriate sub-headings.
3. Regarding the questionnaire that was used, I think an important information that is missing is how to interpret the score. What do high and low scores mean?
4. Did the authors identify any missing data, and how did they manage it?
5. Regarding the discussion, it will be interesting to see the implications of the study findings for the wider healthcare team?
6. Please add a section on the limitations of your study.
Author Response
Dear Reviewer 1,
Thank you very much for the kind comments from you. Please see the attachment.

Reviewer 2 Report
I think this is an important research. Please consider the following points
An abbreviation is a shortened form of a word or phrase; abbreviations of phrases are often composed of the first letter of each word of the phrase (i.e., acronym). To maximize clarity, use abbreviations sparingly. Please write the full name of the abbreviation for the first time.
Line 38: What does ATK stand for?
Line 43: The topic suddenly switches from PCR and ATK testing to Burnout. Please examine the sentence structure and add appropriate sentences.
Line 55: Is that the degree of "engagement”? I believe it needs to be clearly expressed.
Line 64: Please check and correct the wording of the abbreviations PCL and (BDMS).
Line 78: Authors must show the data analysis, including descriptive analysis, exploratory factor analysis for construct validity and factor analysis and Cronbach’s alpha for measuring internal consistency (Please use the table). KMO (Kaiser–Meyer-Olkin) test and Bartlett’s test of sphericity results must determine sample adequacy and appropriateness of data for factor analysis.
Line 98: The estimation of the sample size is a critical topic for the researchers, so it is an indispensable process for obtaining good research results. Thus, authors must indicate the sample size and effect size for t-test, one-way ANOVA, and linear regression analysis.
To conduct power analysis to estimate your sample size, you must write your hypothesis, and based on that you decide what statistical test you will use. It should be one of the inferential statistics. so you should determine the following: alpha, power, effect size.
G*Power provides improved effect size calculators and graphics options, it supports both a distribution-based and a design-based input mode, and it offers five different types of power analyses.
Then, download free programs are to calculate the sample size, such as G. power.
https://www.psychologie.hhu.de/arbeitsgruppen/allgemeine-psychologie-und-arbeitspsychologie/gpower
Table 1: An asterisk is not required when indicating a P-value.
Line 117: Please describe what statistical methods you used. Also, do you describe the statistical methods you used here in your analysis method? Please confirm.
Line 133: Please describe here what statistical methods you used.
Author Response
Dear Reviewer 2,
Thank you very much for the kind comments from you. Please see the attachment.

Round 2
Reviewer 2 Report
I have confirmed that the correction has been made. However, the following points still remain. You can confirm and improve your paper. I believe this research will show a good research outcomes. I will be happy if I can be of any help.
Line 101
2.4 Sample size
You can state below.
A priori analysis for two independent means, one way ANOVA, and linear multiple regression was conducted using the statistical power analysis software: the G * Power (version 3.1) [Faul, F., et al. (2007)]. A sufficient sample size using an alpha 0.05, a power of 0.95, and the required sample size was 1084, 360, and 153, respectively.
Faul, F., Erdfelder, E., Lang, A.-G. and Buchner, A. (2007) G Power 3: A Flexible Statistical Power Analysis Program for the Social, Behavioral, and Biomedical Sciences. Behavior Research Methods, 39, 175-191.
https://doi.org/10.3758/BF03193146
t tests - Means: Difference between two independent means (two groups)
Analysis: A priori: Compute required sample size
Input: Tail(s) = One
Effect size d = 0.2
α err prob = 0.05
Power (1-β err prob) = 0.95
Allocation ratio N2/N1 = 1
Output: Noncentrality parameter δ = 3.2924155
Critical t = 1.6462631
Df = 1082
Sample size group 1 = 542
Sample size group 2 = 542
Total sample size = 1084
Actual power = 0.9500669
In your study, this is an adequate sample size for t-testing.
F tests - ANOVA: Fixed effects, omnibus, one-way
Analysis: A priori: Compute required sample size
Input: Effect size f = 0.25
α err prob = 0.05
Power (1-β err prob) = 0.95
Number of groups = 8
Output: Noncentrality parameter λ = 22.5000000
Critical F = 2.0356185
Numerator df = 7
Denominator df = 352
Total sample size = 360
Actual power = 0.9521702
In your study, this is an adequate sample size for one way ANOVA.
F tests - Linear multiple regression: Fixed model, R² deviation from zero
Analysis: A priori: Compute required sample size
Input: Effect size f² = 0.15
α err prob = 0.05
Power (1-β err prob) = 0.95
Number of predictors = 7
Output: Noncentrality parameter λ = 22.9500000
Critical F = 2.0732820
Numerator df = 7
Denominator df = 145
Total sample size = 153
Actual power = 0.9503254
In your study, this is an adequate sample size for linear multiple regression.
Line 112
You can add the following description after “The final questionnaire comprised 40 questions.”
For analyzing the questionnaire items, the Kaiser-Meyer-Olkin (KMO) test and Bartlett’s test of Sphericity were applied for sampling adequacy. The KMO value of 0.5 and above indicated that the analysis could proceed to Exploratory Factor Analysis.
Line 116
Why is there a percentage indication in Table one?
The table should be separated for patient characteristics and t-test and ANOVA results.
Since you are conducting a t-test and ANOVA, the values that need to be shown here are the mean and standard deviation, t-value or F-value, and post hoc test values for ANOVA.
For items where ANOVA showed significant differences, post hoc values should be indicated. In this way, the relationship to linear multiple regression analysis can be expressed more clearly.
Line 200
Please revise from (P-value =0931) to (P-value =0.931).
You don't need an asterisk in Table 3.
Line 202
Please indicate here the statistical methods.
Author Response
Dear Editor and Reviewers
We wish to submit a revision manuscript entitled “Physician Engagement Before and During The COVID-19 Pandemic in Thailand” for consideration. We confirm that the correction have been made. We truly appreciate all comments from the reviewers. Please consider the revised manuscript with track changes and the following point-by-point responses in attached file.
Best Regards.
